# Evaluation of ^60^Co Irradiation on Volatile Components of Turmeric (Curcumae Longae Rhizoma) Volatile Oil with GC–IMS

**DOI:** 10.3390/foods12132489

**Published:** 2023-06-26

**Authors:** Ye He, Lu Yin, Wei Zhou, Hongyan Wan, Chang Lei, Shunxiang Li, Dan Huang

**Affiliations:** 1State Key Laboratory of Chinese Medicine Powder and Medicine Innovation in Hunan (Incubation), Science and Technology Innovation Center, Hunan University of Chinese Medicine, Changsha 410208, China; hy070915@126.com (Y.H.); yl024083@163.com (L.Y.); fxxx99w@163.com (W.Z.); wanhongyan2001@163.com (H.W.); leichang1231@126.com (C.L.); 2Hunan Engineering Technology Research Center for Bioactive Substance Discovery of Chinese Medicine, School of Pharmacy, Hunan University of Chinese Medicine, Changsha 410208, China; 3Hunan Province Sino-US International Joint Research Center for Therapeutic Drugs of Senile Degenerative Diseases, Changsha 410208, China

**Keywords:** Curcumae Longae Rhizoma, volatile oil, ^60^Co, GC–IMS

## Abstract

^60^Co irradiation is an efficient and rapid sterilization method. The aim of this work is to determine the changes in essential-oil composition under different irradiation intensities of ^60^Co and to select an appropriate irradiation dose with GC–IMS. Dosages of 0, 5, and 10 kGy of ^60^Co were used to analyze turmeric (Curcumae Longae Rhizoma) volatile oil after ^60^Co irradiation (named JH-1, JH-2, and JH-3). The odor fingerprints of volatile organic compounds in different turmeric volatile oil samples were constructed by headspace solid-phase microextraction and gas chromatography–ion mobility spectrometry (GC–IMS) after irradiation. The differences in odor fingerprints of volatile organic compounds (VOCs) were compared by principal component analysis (PCA). The results showed that 97 volatile components were detected in the volatile oil of Curcuma longa, and 64 components were identified by database retrieval. With the change in irradiation intensity, the volatile compounds in the three turmeric volatile oil samples were similar, but the peak intensity was significantly different, which was attributed to the change in compound composition and content caused by different irradiation doses. In addition, the principal component analysis showed that JH-2 and JH-3 were relatively correlated, while JH-1 and JH-3 were far from each other. In general, different doses of ^60^Co irradiation can affect the content of volatile substances in turmeric volatile oil. With the increase in irradiation dose, the peak area decreased, and so the irradiation dose of 5 kGy/min was better. It is shown that irradiation technology has good application prospects in the sterilization of foods with volatile components. However, we must pay attention to the changes in radiation dose and chemical composition.

## 1. Introduction

Curcumae Longae Rhizoma, a rootstock plant called turmeric belonging to Fam. Zingiberaceae in China, is extensively cultivated in South China, such as in the Sichuan, Guangxi, Guangdong, Yunnan, and Zhejiang provinces. Turmeric has been widely used in traditional Chinese medicine to promote Qi circulation, dissolve blood sludge, induce menstruation, and relieve pain. It can be used to relieve pain in the chest and hypochondriac regions, as well as treat amenorrhea, mass formation in the abdomen, rheumatic pain of the shoulders and arms, traumatic swelling, and pain (Committee for the Pharmacopoeia of P.R. China, 2020). Modern research shows that the abundant volatile oil and curcumin in turmeric (up to about 4%) have anti-tumour, anti-inflammatory, and antioxidant effects [1].

The quality of traditional Chinese medicine will have an impact on its efficacy. With the development of modern medicine, the quality standards of traditional Chinese medicine are also constantly improving. However, the source of traditional Chinese medicine is complex, and it is easy to be contaminated by bacteria in the process of production and storage. Traditional dry-heat sterilization, damp-heat sterilization and ethylene oxide gas sterilization technologies easily destroy the effective components of drugs, affecting the efficacy or residual organic solvents. The sterilization operation should remove microorganisms and ensure the quality and efficacy of drugs [2]. ^60^Co-γ irradiation ray sterilization method, widely used in the sterilization of Chinese medicinal materials, is a new disinfection and sterilization process in the 20th century, which is based on the high-energy rays produced by X-ray, γ ray, and other ionizing radiation to inhibit the continuation of pests and diseases and to effectively kill insects. Sterilization at room temperature and high efficiency, with simple operation and other characteristics, dosages of 5, and 10 kGy are most widely used [3,4]. However, the influence of the irradiation dose on the component of traditional Chinese medicine remains unknown. Therefore, it is of great significance to explore the sterilization dose of irradiation sterilization of different traditional Chinese medicine varieties using ^60^Co, which can be of great reference value to the quality problems of traditional Chinese medicine and its preparation products. 

The traditional detection method of volatile compounds in turmeric uses gas chromatography-mass spectrometry (GC–MS) [5], but it takes a long time and has low sensitivity, and GC-MS needs complicated pre-treatments and is constrained in distinguishing isomeric molecules [6]. In recent years, a novel and powerful device, gas chromatography (GC) coupled with ion mobility spectrometry (IMS), has been chosen to accurately test flavor compounds, especially the volatile oil components in various foods. Gas chromatography–ion mobility spectrometry (GC–IMS) which has the advantages of simple operation, strong separation ability, short detection cycle, and retaining the original flavor of samples to the greatest extent [7], has been successfully applied to food [8], biological and aquatic odor analyses [9], quality detection [10], and other fields. However, there is no report on the study of volatile organic compounds in turmeric volatile oil by GC–IMS technology.

In this study, the volatile oil of turmeric was extracted by steam distillation and irradiated, with ^60^Co rays of different intensities (Dosages of 0, 5, and 10 kGy). A total of 64 components were detected using GC–IMS technology to compare the changes before and after irradiation. Finally, the composition changes of volatile oil under different irradiation intensities were given to provide a certain reference for the production and sterilization of turmeric and its preparations. This study provides a sound basis for the use of ^60^Co-γ ray irradiation sterilization technology during the preparation of medicinal herbs. GC–IMS, which has the advantages of simple operation, strong separation ability, short detection cycle, and retaining the original flavor of samples to the greatest extent, can be successfully applied to foods.

## 2. Materials and Methods

### 2.1. Materials

Turmeric was collected from Baise, Guangxi Province, China, and identified by Prof. Zhaoming Xie at the Hunan Academy of Traditional Chinese Medicine. A voucher specimen (HNATCM2022-006) was deposited in the herbarium of the Hunan Academy of Traditional Chinese Medicine.

### 2.2. Isolation of the Essential Oils

The essential oil was extracted through steam distillation by referring to the Chinese Pharmacopoeia 2020 edition (part 4) volatile oil determination method (General Principle 2204) for determination. We added 500 g of dried turmeric with an appropriate amount of water and a few glass beads into a 1000 mL round-bottom flask, treating the solution by slowly heating it to a boil, followed by keeping it slightly boiling for 6 h. The upper oil phase (the crude essential oil of turmeric) was gathered and sealed and stored away from light at 4 °C for further use.

### 2.3. Extraction Yield

After extraction, the extraction rate of volatile oil is calculated according to the following formula:Extraction yield (%) = m_1_/m_0_ × 100(1)
where m_1_ is the total mass of the extracted oil, and m_0_ is the initial mass of the turmeric used in each extraction.

### 2.4. ^60^Co-γ Irradiation

The resulting curcumin EOs was dehydrated with anhydrous Na_2_SO_4_ and then divided into three equal parts for ^60^Co irradiation. The dose rates were 0, 5, and 10 kGy/min. The ^60^Co γ radiation source was located at Hunan Radiological Technology Application Research Center (Changsha, China). 

### 2.5. Analysis by GC–IMS

In the experiment, the volatiles were concentrated and separated by headspace solid-phase microextraction, with reference to Wang [11] and other methods and appropriate adjustments. Precisely-measured 50 μL of turmeric volatile oil sample was transferred into a 20 mL headspace bottle with Teflon spacer seal. The headspace bottle was heated at 80 °C and incubated for 10 min at 500 RPM. Then 100 μL of the sample was injected in non-shunt mode, and the temperature of the injection needle was kept at 85 °C.

The components of volatile compounds were identified by chromatography–ion mobility spectroscopy (GC–IMS; FlavourSpec^®^, G.A.S., Berlin, Germany). Gas chromatography (GC) was performed under the following conditions: carrier gas, nitrogen (99.99%); column, mxt-5 (15.0 m length × 0.53 mm ID × 1 μm thickness); running time, 50 min; flow rate, initial 2.0 mL/min, holding for 2 min, linearly increasing to 100 mL/min within 18 min, and holding for 20 min. Ion mobility spectroscopy (IMS) was carried out under the following conditions: drift gas, nitrogen (99.99%); flow rate, 150 mL/min; IMS detector temperature, 45 °C.

Three parallel samples are set for each irradiation intensity for volatile oil, and the difference in the spectrum of volatile organic compounds in the sample can be given after analysis. The NIST database and IMS database built into the software can conduct a qualitative analysis of substances.

### 2.6. Statistical Analysis 

The analysis software Vocal matched with the instrument is used to view the qualitative and quantitative analysis spectrum and data. The NIST database and IMS database built into the application software can be used for qualitative analysis of substances. The porter plug-in directly compares the spectrum differences between samples (three-dimensional spectrum, two-dimensional top view, and difference spectrum). We compared the fingerprint of the gallery plot plug-in to intuitively and quantitatively compare the differences in volatile organic compounds between different samples. A dynamic PCA plug-in was used for dynamic principal component analysis, cluster analysis of samples, and rapid determination of unknown and unknown samples.

## 3. Results

### 3.1. Flavor Differences in Turmeric Volatile Oil under Different Irradiation Intensities Detected by GC–IMS

Three-dimensional topographic plots of turmeric irradiated by three different doses are shown in Figure 1A. We assessed the volatile organic compounds (VOC) in different samples from three perspectives: retention time, drift time, and peak intensity. The number of VOCs and the signal intensity of the peaks differed slightly among the three samples. With the increase in irradiation intensity, only the signal intensity of the peaks changes slightly, and almost no new compounds form.

A two-dimensional topographic spectrum (planform of the 3D plot) was also obtained for its difficulty to observe the differences between three-dimensional groups (Figure 1B). In this plot, the red vertical line at 1.0 on the left is the reaction ion peak (RIP), and the background image is blue. Each point on both sides of the reaction ion peak represents a VOC, the color depth represents the volatile-compound content, the white area represents the low compound content, and the red area represents the high compound content. Using the difference comparison mode, we select the spectrum (JH-1) of one sample as the reference, and we deduct the spectrum of other samples from the reference to obtain Figure 1C. If the volatile organic compounds of the two samples are consistent, then the background after deduction is white, while red signifies that the concentration of the substance is higher than the reference, and blue implies that the concentration of the substance is lower than the reference. It can be seen from Figure 1B,C that signals are concentrated in areas A, B, and C. The color of some compounds in areas A and C is deepened, and the color of compounds in area B is lighter, which suggests that with the increase in irradiation dose, the compound content in regions A and C increases and in the regioqualitative analysis of volatile organic compounds.

Figure 1D shows the binary spectra of all volatile substances of turmeric volatile oil under three irradiation intensities. The volatile compounds in the volatile oil of turmeric were analyzed by GC. The NIST database and IMS database built into the IMS library were according to the retention index, retention time, and ion migration time. We identified 64 volatile components shown in Table 1 from the three sample varieties in Figure 1D. The substance numbers in Figure 1D are consistent with those in Table 1. 

According to Table 1 and Table 2, there are 64 monomers and dimers of volatile substances identified in turmeric volatile oil, including 17 alcohols and phenols, 11 aldehydes, 11 ketones, 9 terpenes, 8 esters, 4 carboxylic acids and their derivatives, 3 furans, and 1 thiophene. The chemical formula of the identified monomer and dimer is the same as the CAS number, but the form is different. The results are shown in Table 1 and Table 2.

### 3.2. Odor Fingerprint of Volatile Substances

The fingerprint of VOCs corresponding to each sample with different irradiation intensities is displayed in Figure 1E. The same row represents the signal peaks of volatile compounds in the same turmeric volatile oil sample, and the same column represents the signal peaks of the same volatile compound in different turmeric volatile oil samples. The color from light to dark indicates the content of volatile compounds from low to high. The fingerprints of volatile organic compounds collected from three kinds of samples were divided into three regions with different colors. The volatile organic compounds in turmeric volatile oil varied substantially with different irradiation intensities. 

The substances in the green box in Figure 1E, such as citronellol, alpha terpineol, decanoic acid, methyl decanoate, and linalool-m, have the highest content in JH-1 sample and gradually decrease with the increase in irradiation dose. The content of substances in the yellow area is the highest in JH-2 samples, such as 2,5-dimethylfuran, acetate, and (E)-2-pentanal. The content of substances in the red area is the highest in the JH-3 sample and the lowest in the JH-1 sample, such as 2,5-dimethylfuran, acetal, and (E)-2-glutaraldehyde. The JH-2 content of Yellow Zone 2 is the lowest among the three samples, such as 2-acetylfuran, benzaldehyde-d, benzaldehyde-m, furfural-d, furfural-m, and 2-methylbutanoic acid.

### 3.3. Similarity Analysis

Figure 1F is the PCA diagram of three volatile oil samples at different irradiation intensities, which can visually show the differences between different products. The volatile organic compounds in JH-1 and JH-2 are similar, and the distance between them is very close. The gap between JH-1 and JH-3 is the largest, and the distance between them is the farthest.

Figure 1G shows the fingerprint similarity analysis of three volatile oil samples with different irradiation intensities. It can also be seen from the figure that the volatile organic compounds of JH-2 and JH-3 are very similar, the distance between them is very close, and the results of the principal component analysis are consistent with those of fingerprint analysis. There are substantial differences in volatile organic compounds between volatile oil samples (JH-1, JH-3) with large differences in irradiation intensity.

## 4. Discussion

The main active components and volatile components of turmeric volatile oil exposed to different doses of ^60^Co-γ irradiation were determined and analyzed. By using headspace sampling and GC–IMS technology, the compounds can be qualitatively analyzed according to the GC retention time and ion migration time of volatile substances. A total of 64 volatile compounds were identified by GC–IMS analysis and built-in NIST database retrieval. The results showed that the contents of various volatile substances were different under different irradiation intensities. The volatile oil in turmeric is responsible for the aroma of turmeric, while curcumin (curcumin and its analogues) is responsible for its bright yellow color [12,13]. Some literature studies on turmeric butter have identified sesquiterpenoids and monoterpenoids as the main components [14], including gingerone, curcumene, curcumin, sabinene, borneol, caryophyllene, and other compounds [15]. The results of this study show that ethyl-2-phenylacetate, 2-ethylfuran, 2-butanone, 1-pentanol, 2-methylbutanoic acid, 3-hydroxy-2-butanone, linalool-d, eugenol, ethyl propanoate, ethyl pentanoate, and diethyl succeed are the lowest in JH-1 and the highest in JH-3. 2,5-Dimethylfuran, acetal, (E)-2-pentanal, and three unmatched compounds have the highest content in JH-2, followed by JH-1, and JH-3. The contents of 2-methyl-1-butanol, 2-methylpropanal, 2,3-butanedione, 2-nonanone, 2,5-dimethylthiophene, 2-furanmethanol, beta-ocimene, linalool-m, citronellol, alpha-terpineol, decanoic acid, and methyl decanoate decreased gradually from JH-1 to JH-3. The content of 2-acetylfuran is higher in JH-3, followed by JH-1; and almost none is present in JH-2. The content of 2-propanol in JH-1 was the highest, JH-3 was the second, and JH-2 was the lowest. The rest of the ingredients did not change significantly. According to the literature, turmerin, turmerone, elemene, furanodiene, curdione, bisacurone, cyclocurcumin, calebin A, and germacrone and other compounds in turmeric volatile oil have anti-inflammatory and anti-cancer activities [16,17,18], anti-hyperlipidemic property [19,20,21], as well as used in the prevention of asthma [22], treatment of respiratory diseases, and anti-oxidation in vitro effect [23,24]. Most of these main components did not change much because of the influence of irradiation, and so ^60^Co-γ ray irradiation did not have a great impact on the effectiveness of turmeric volatile oil.

Dosages of 0, 5, and 10 kGy of ^60^Co were used to analyze turmeric (Curcumae Longae Rhizoma) volatile oil after ^60^Co Irradiation (named JH-1, JH-2 and JH-3). With the increase in irradiation dose, the peak area decreased. It is of great significance to explore the sterilization dose of irradiation sterilization of different traditional Chinese medicine varieties using ^60^Co. Dosages of 5 and 10 kGy are the most widely used. There is a maximum level and limit for radiation which should be discussed in the future. It could be better to do it for 0, 2.5, 5, 7.5, and 10.

In this study, the author found that gas chromatography–mass spectrometry can effectively identify volatile odor compounds such as alcohols, ketones, aldehydes, esters, and terpenes. Food [25,26], agriculture [27,28,29], and traditional Chinese medicine field are widely used [30]. Gas-phase ion mobility spectrometry widely used in food [25,26], agriculture [27,28,29], and traditional Chinese medicine fields can quickly and accurately conduct a qualitative analysis of turmeric volatile oil under different irradiation doses and elucidate the differences in the odor of volatile organic compounds between samples [30]. However, there are still some limitations. The author has not conducted a further quantitative analysis of each compound. The next step will be to do a more explicit quantitative analysis of the specific components of the volatile oil.

## 5. Conclusions

In this study, the volatile components of turmeric volatile oil samples with three different irradiation intensities were analyzed by GC–IMS. A total of 97 volatile substances were detected, and a total of 64 components were determined by database retrieval, such as dimers of some substances. The volatile organic compounds in three turmeric volatile oil samples, mainly including terpenes, esters, aldehydes, alcohols, and ketones, were found. The results showed that the chemical components of the three turmeric volatile oil samples were similar, but the contents were quite different, suggesting that the irradiation intensities might have an impact on the volatile organic compounds of turmeric volatile oil. JH-1 and JH-3 have great differences in irradiation strength, and the difference in the principal component analysis is also large, indicating that the difference in chemical composition is the largest. This study provides a scientific basis for the dose control of irradiation sterilization of turmeric and its volatile oil. With the increase in irradiation dose, the peak area decreased, and so the irradiation dose of 5 kGy/min was better for ^60^Co irradiation of turmeric (Curcumae Longae Rhizoma). This study provides a sound basis for the use of ^60^Co-γ ray irradiation sterilization technology during the preparation of medicinal herbs.

It is shown that irradiation technology has good application prospects in the sterilization of foods with volatile components, However, attention must be paid to the changes in radiation dose and chemical composition. GC–IMS which has the advantages of simple operation, strong separation ability, short detection cycle, and retaining the original flavor of samples to the greatest extent, can be successfully applied to foods.

## Figures and Tables

**Figure 1 foods-12-02489-f001:**
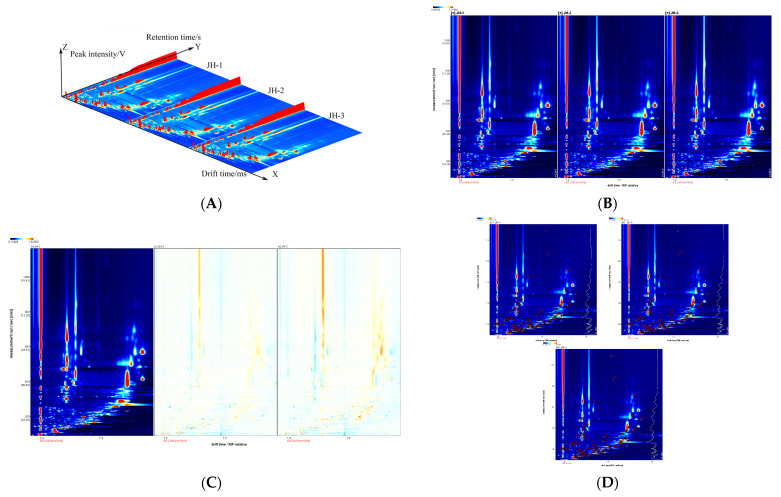
Volatile compounds analysis by GC-IMS. (**A**) 3D-topographic; (**B**) topographic plots; (**C**) topographic subtraction plots; (**D**) post-irradiation topographic plots; (**E**) volatile compounds fingerprint comparisons. Each row represents all the signals selected in a sample. Each column represents the signal of the same volatile compound. (**F**) Principal component analysis; (**G**) Fingerprint similarity analysis.

**Table 1 foods-12-02489-t001:** Volatile substances contained in turmeric volatile oil.

No	Compounds	Molecular Formula	RI	Rt/s	Dt/ms (RIPrel)
1	Methyl decanoate	C_11_H_22_O_2_	1508.0	1079.39	1.54878
2	Decanoic acid	C_10_H_20_O_2_	1360.2	867.201	1.57124
3	Eugenol	C_10_H_12_O_2_	1341.6	840.393	1.29711
4	alpha-Terpineol	C_10_H_18_O	1206.8	646.87	1.22248
5	Diethyl succinate	C_8_H_14_O_4_	1234.7	686.937	1.29422
6	Citronellol	C_10_H_20_O	1197.5	633.591	1.34963
7	Linalool	C_10_H_18_O	1106.9	503.438	1.2229
8	2-Nonanone	C_9_H_18_O	1091.9	481.9	1.88944
9	1,8-Cineole	C_10_H_18_O	1027.6	389.61	1.73145
10	beta-Ocimene	C_10_H_16_	1054.5	428.22	1.21822
11	Benzeneacetaldehyde	C_8_H_8_O	1045.1	414.765	1.2491
12	Limonene	C_10_H_16_	1025.6	386.685	1.20792
13	alpha-Terpinene	C_10_H_16_	1015.0	371.475	1.22263
14	6-Methyl-5-hepten-2-one	C_8_H_14_O	990.5	339.885	1.17704
15	beta-Pinene	C_10_H_16_	976.0	327.6	1.21969
16	beta-Pinene	C_10_H_16_	976.0	327.6	1.64175
17	Camphene	C_10_H_16_	944.9	301.275	1.21822
18	alpha-Pinene	C_10_H_16_	931.8	290.16	1.22116
19	alpha-Pinene	C_10_H_16_	931.1	289.575	1.67557
20	2-Ethylhexanol	C_8_H_18_O	1018.3	376.21	1.79856
21	2-Octanone	C_8_H_16_O	999.2	348.832	1.7651
22	Benzaldehyde	C_7_H_6_O	959.8	313.914	1.15225
23	Benzaldehyde	C_7_H_6_O	959.4	313.517	1.47206
24	Ethyl pentanoate	C_7_H_14_O_2_	901.2	264.315	1.71158
25	2-Heptanone	C_7_H_14_O	891.6	256.379	1.62862
26	2-Furanmethanol	C_5_H_6_O_2_	873.3	246.856	1.1188
27	2-Acetylfuran	C_6_H_6_O_2_	911.1	272.648	1.45333
28	2,5-Dimethylthiophene	C_6_H_8_S	856.5	238.127	1.07866
29	Furfural	C_5_H_4_O_2_	828.3	223.446	1.08401
30	Furfural	C_5_H_4_O_2_	826.7	222.652	1.3329
31	2-Methylbutanoic acid	C_5_H_10_O_2_	827.5	223.049	1.46938
32	2-Hexanol	C_6_H_14_O	793.1	205.193	1.56974
33	2-Hexanone	C_6_H_12_O	791.9	204.546	1.50912
34	2,3-Butanediol	C_4_H_10_O_2_	778.3	198.14	1.3627
35	1-Pentanol	C_5_H_12_O	760.5	190.908	1.25055
36	Acetal	C_6_H_14_O_2_	748.9	186.155	0.96913
37	(E)-2-Pentenal	C_5_H_8_O	747.3	185.535	1.36062
38	3-Hydroxy-2-butanone	C_4_H_8_O_2_	733.1	179.749	1.33778
39	2,5-Dimethylfuran	C_6_H_8_O	739.9	182.503	1.0218
40	Methyl butanoate	C_5_H_10_O_2_	716.5	172.993	1.15331
41	Ethyl propanoate	C_5_H_10_O_2_	695.4	164.398	1.44128
42	2-Pentanone	C_5_H_10_O	686.6	161.106	1.37552
43	2-Ethylfuran	C_6_H_8_O	674.9	157.998	1.3211
44	3-Methylbutanal	C_5_H_10_O	650.6	151.597	1.4118
45	1-Butanol	C_4_H_10_O	656.2	153.06	1.37552
46	3-Methylbutanal	C_5_H_10_O	645.1	150.134	1.20319
47	2-Butanone	C_4_H_8_O	591.7	136.053	1.24854
48	2-Methyl propanal	C_4_H_8_O	555.1	126.36	1.28596
49	2-Propanol	C_3_H_8_O	565.4	129.104	1.22814
50	2,3-Butanedione	C_4_H_6_O_2_	582.1	133.493	1.18732
51	Acetic acid	C_2_H_4_O_2_	565.4	129.104	1.16918
52	2-Propanone	C_3_H_6_O	539.2	112.462	1.11816
53	Ethanol	C_2_H_6_O	1508	105.513	1.12723
54	Methyl acetate	C_3_H_6_O_2_	1360.2	125.263	1.20319
55	Pentanal	C_5_H_10_O	1341.6	163.667	1.18165
56	Ethyl 2-phenylacetate	C_10_H_12_O_2_	1206.8	692.941	1.78609
57	Acetophenone	C_8_H_8_O	1234.7	471.206	1.19245
58	Linalool	C_10_H_18_O	1197.5	503.498	1.76126
59	alpha-Phellandrene	C_10_H_16_	1106.9	356.925	1.68764
60	Tricyclene	C_10_H_16_	1091.9	283.788	1.67365
61	(E)-2-Hexenol	C_6_H_12_O	1027.6	234.035	1.15052
62	1-Hexanol	C_6_H_14_O	1054.5	232.537	1.63849
63	Ethyl butanoate	C_6_H_12_O_2_	1045.1	216.141	1.57005
64	2-Methyl-1-butanol	C_5_H_12_O	1025.6	178.89	1.48673

**Table 2 foods-12-02489-t002:** Area of turmeric volatile oil.

No	Compounds	Molecular Formula	[+] JH-1	[+] JH-2	[+] JH-3
1	Methyl decanoate	C_11_H_22_O_2_	853.04	807.50	706.77
2	Decanoic acid	C_10_H_20_O_2_	343.61	425.17	323.73
3	Eugenol	C_10_H_12_O_2_	3892.53	4916.40	5396.33
4	alpha-Terpineol	C_10_H_18_O	19,657.51	18,387.17	17,492.96
5	Diethyl succinate	C_8_H_14_O_4_	4842.63	5279.30	5497.18
6	Citronellol	C_10_H_20_O	806.72	785.28	774.24
7	Linalool	C_10_H_18_O	8368.12	8339.72	8003.81
8	2-Nonanone	C_9_H_18_O	1129.50	845.90	803.89
9	1,8-Cineole	C_10_H_18_O	17,383.17	17,589.74	17,990.91
10	beta-Ocimene	C_10_H_16_	730.79	652.68	599.08
11	Benzeneacetaldehyde	C_8_H_8_O	461.94	443.61	419.23
12	Limonene	C_10_H_16_	1525.97	1546.91	1559.40
13	alpha-Terpinene	C_10_H_16_	3982.81	3847.04	3737.19
14	6-Methyl-5-hepten-2-one	C_8_H_14_O	764.36	841.66	779.81
15	beta-Pinene	C_10_H_16_	1892.73	1854.60	1873.25
16	beta-Pinene	C_10_H_16_	6168.36	6338.65	6214.42
17	Camphene	C_10_H_16_	5470.41	5424.69	5375.12
18	alpha-Pinene	C_10_H_16_	1211.24	1190.02	1129.20
19	alpha-Pinene	C_10_H_16_	6032.87	6101.62	6333.75
20	2-Ethylhexanol	C_8_H_18_O	196.51	191.32	202.42
21	2-Octanone	C_8_H_16_O	734.45	737.80	727.27
22	Benzaldehyde	C_7_H_6_O	248.36	256.24	260.37
23	Benzaldehyde	C_7_H_6_O	302.21	255.45	291.10
24	Ethyl pentanoate	C_7_H_14_O_2_	16,363.09	16,853.35	16,297.10
25	2-Heptanone	C_7_H_14_O	6768.49	6831.28	6604.10
26	2-Furanmethanol	C_5_H_6_O_2_	693.97	694.50	633.27
27	2-Acetylfuran	C_6_H_6_O_2_	43.49	19.02	56.13
28	2,5-Dimethylthiophene	C_6_H_8_S	355.34	398.57	404.40
29	Furfural	C_5_H_4_O_2_	151.02	163.77	172.80
30	Furfural	C_5_H_4_O_2_	148.19	162.08	192.78
31	2-Methylbutanoic acid	C_5_H_10_O_2_	97.08	98.83	103.59
32	2-Hexanol	C_6_H_14_O	1552.99	1696.63	1741.51
33	2-Hexanone	C_6_H_12_O	702.91	717.69	727.46
34	2,3-Butanediol	C_4_H_10_O_2_	2616.14	2590.83	2552.24
35	1-Pentanol	C_5_H_12_O	98.02	97.41	107.12
36	Acetal	C_6_H_14_O_2_	43.00	48.17	45.78
37	(E)-2-Pentenal	C_5_H_8_O	430.50	467.99	387.28
38	3-Hydroxy-2-butanone	C_4_H_8_O_2_	794.53	799.28	854.01
39	2,5-Dimethylfuran	C_6_H_8_O	125.36	141.91	111.45
40	Methyl butanoate	C_5_H_10_O_2_	89.02	93.33	91.42
41	Ethyl propanoate	C_5_H_10_O_2_	2091.53	2078.08	2150.97
42	2-Pentanone	C_5_H_10_O	508.33	528.88	573.27
43	2-Ethylfuran	C_6_H_8_O	1519.15	1480.31	1638.41
44	3-Methylbutanal	C_5_H_10_O	149.00	147.46	136.26
45	1-Butanol	C_4_H_10_O	2718.89	2744.27	2798.98
46	3-Methylbutanal	C_5_H_10_O	786.67	816.78	863.78
47	2-Butanone	C_4_H_8_O	1163.04	1258.54	1331.09
48	2-Methyl propanal	C_4_H_8_O	116.66	127.12	118.02
49	2-Propanol	C_3_H_8_O	538.81	443.21	534.13
50	2,3-Butanedione	C_4_H_6_O_2_	715.88	646.28	653.28
51	Acetic acid	C_2_H_4_O_2_	899.26	900.87	978.94
52	2-Propanone	C_3_H_6_O	11,632.68	12,363.82	12,586.06
53	Ethanol	C_2_H_6_O	2364.98	2789.25	2848.56
54	Methyl acetate	C_3_H_6_O_2_	355.35	368.09	363.62
55	Pentanal	C_5_H_10_O	59.13	56.64	56.97
56	Ethyl 2-phenylacetate	C_10_H_12_O_2_	3291.02	3610.35	3732.50
57	Acetophenone	C_8_H_8_O	6600.73	6488.00	6267.98
58	Linalool	C_10_H_18_O	6085.73	6678.48	7028.94
59	alpha-Phellandrene	C_10_H_16_	1909.42	1986.81	1925.54
60	Tricyclene	C_10_H_16_	4633.64	4665.30	4772.88
61	(E)-2-Hexenol	C_6_H_12_O	623.21	621.26	613.55
62	1-Hexanol	C_6_H_14_O	330.60	337.76	336.93
63	Ethyl butanoate	C_6_H_12_O_2_	267.79	278.89	273.73
64	2-Methyl-1-butanol	C_5_H_12_O	318.35	309.08	301.05

## Data Availability

Data is contained within the article.

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
