# Peer review of "Evaluation of 60Co Irradiation on Volatile Components of Turmeric (Curcumae Longae Rhizoma) Volatile Oil with GC–IMS"

_foods, 2023, doi:10.3390/foods12132489_

Round 1

Reviewer 1 Report

GC-IMS is a promising and sensitive technique for the analysis of volatile compunds to be found in samples. This manuscript # Foods-2456771 deals with the analysis of volatile compounds in turmeric volatile oil samples with three different irridation intensities. A total of 97 volatile substances was determined while a total of 64 components were determined by database retrieval. The results are looking promising.  I would recommend its publication in the journal. Before publication, several points should be addressed as follows;

1) in Introduction section

-More information should be given regarding  GC-IMS technique and the comparing with GC-MS would be better.

-No information was given regarding the analysis of this matrix with different analytical technique 

-More information should be given in realtion to the aspect of green metahdology of this technique 

2) Section 2.5

Did the authors try to use more narrow i.d. column for the analysis? The use of narrow i.d. column may allow the detection of more volatile compounds.

3) Results 

More IMS data should be given

English needs minor revisions

Reviewer 2 Report

In this manuscript 60Co irradiation was used as a sterilization method, to determine the changes in essential-oil composition of turmeric under different irradiation intensities. It is an interesting research, but the performance quality is low and needs very serious revision.

In whole manuscript, it should be a constant way to write the tumeric, it was written with different ways and names.

Abstract should be informative and presenting some data and numbers. It should be difined what is JH-2 and JH-3 or JH-1 and JH-3?

From abstract, it is understood that the effect of radiation on tumeric was investigated on the changes in the essential oils. This should be clarified that effect of radiation was studied on the essential oil itself after extraction from tumeric.  The importance and neccesity of radiation of essential oil instead of tumeric itself should be discussed in Introduction part as well.

Actually, the main issue in this study is defining the effect of irradiation by the peak area. Peak area can be different by the amount and volume of the sample injected to the GC even for one given sample. So, how it can be used to compare the effect of radiation for different samples and in different dosages?

There is a maximum level and limit for radiation which should be discussed in the text. It could be better to do it for 0, 2.5, 5, 7.5 and 10.

It was very valuable to use an internal standard prior to GC analysis to quantify the each essential oil components amount.

Choromatogram of the GC of essential oil should be added in the manuscript.

No data reported for the level and percentage of the essential oil after radiation with different dosage. One Table for changes in the peack area of different essential oils could be better.

Abbreviations should be defined. Using "we" or "the author found" is not scientific and should be revised. Paragraph separation for different subjects in the discussion part should be done.

Round 2

Reviewer 1 Report

The authors addressed all my concern and it is now suitable for publication.

minor modifications are needed.

Reviewer 2 Report

Comments and suggestion are included in the revised manuscript, therefore its acceptance is suggested.